# Deep Dynamic AutoEncoder for Vision BERT Pretraining

## Abstract

Recently, masked image modeling (MIM) has demonstrated promising prospects in self-supervised representation learning. However, existing MIM frameworks recover all masked patches equivalently, ignoring that the reconstruction difficulty of different patches can vary sharply due to their diverse distance from visible patches. In this paper, we propose Deep Dynamic AutoEncoder (DDAE), a novel MIM framework that dynamically focuses on patch reconstructions with different degrees of difficulty at different pretraining phases and depths of the model. In addition to raw pixel regression, DDAE performs dynamic feature self-distillation for intermediate layers to learn semantic information. Our methodology provides more locality inductive bias for ViTs, especially in deep layers, which inherently makes up for the absence of local prior for self-attention mechanism. Moreover, our core design deep dynamic supervision can be migrated into existing MIM methods (*e.g.*, MAE, BEiT-v2) seamlessly. The experimental results demonstrate the effectiveness of our approach. As a tokenizer-free framework, the base-size DDAE can achieve 83.5% top-1 accuracy with only 100 epochs pretraining, surpassing MAE and BEiT pretrained for 800 epochs. For a longer pretraining schedule, DDAE achieves 84.3% top-1 accuracy on Imagenet-1K, and 49.3% mIoU on ADE20K for semantic segmentation.

## 1 Introduction

Aided by the rapid gains in hardware, deep learning has ushered in the era of big models and big data. Along with the ever-growing model capacity, the demand for data can easily reach hundreds of millions (Dosovitskiy et al., 2020), which is not publicly accessible for labeled data. Self-Supervised Learning(SSL) frameworks, such as DINO (Caron et al., 2021), MOCO (Chen et al., 2021), BeiT (Bao et al., 2021), etc., have grown in concern in vision model pretraining without the need for labels. In particular, the recently proposed Masked Image Modeling(MIM) methods (He et al., 2022; Xie et al., 2022b; Dong et al., 2021; Wei et al., 2022b; Chen et al., 2022b; Dong et al., 2022; Chen et al., 2022c) have shown remarkably impressive performance in a variety of vision tasks, demonstrating the promise to unify computer vision and natural language processing(NLP) pretraining (Peng et al., 2022a).

Inspired by BERT (Devlin et al., 2018) in NLP, MIM pretrains the encoder by reconstructing the masked image patches from visible patches. Existing MIM methods can be divided into two categories according to the need for an additional tokenizer. **Two-stage methods:** Represented by the pioneer work BEiT (Bao et al., 2021), two-stage methods firstly transform image patches into semantic visual tokens through a pre-trained tokenizer, then reconstruct the corresponding tokens of masked image patches to pretrain the encoder. Tokenizer needs to be offline pretrained with fixed model architectures and extra data (Zhang et al., 2019b; Ramesh et al., 2021; Radford et al., 2021), some methods further require an off-the-shelf DNN as teacher to distill tokenizer pre-training (Peng et al., 2022a). **One-stage methods:** Taking the recent work MAE (He et al., 2022) as a representative, MAE constructs an asymmetric encoder-decoder structure, and directly performs raw pixel regression of masked image patches. One-stage methods are tokenizer-free, the reconstruction target includes raw pixels, features (*e.g.*, HOG) and self-distillation targets (He et al., 2022; Xie et al., 2022b; Wei et al., 2022a; Chen et al., 2022c). MIM methods enable Vision Transformers(ViTs)to learn rich visual representations and show great potential in various downstream tasks.

Naturally, we ask why MIM works. Facilitated by self-attention mechanism, ViTs excel at modeling long-range dependencies, but different from CNN, ViTs' lack of local prior(*i.e.*, a pixel is more related to its neighbors than the distant pixels) largely leads to its need for large dataset pretraining (*e.g.*, JFT-300M) or special training techniques (*e.g.*, a DeiT-style distillation method (Touvron et al., 2021b)). Therefore, we conjecture that the success of MIM comes from the fine-grained reconstruction of masked image patches. The generation task built by MIM enables ViTs to pay more attention to vicinity besides using global semantic reasoning. Thus, it inherently makes up for the absence of local prior in the structure of ViTs, which is essentially different from the discriminative task in supervised pretraining or contrastive learning.

However, success comes with remaining obstacles. At present, all MIM frameworks recover all patches equivalently, ignoring the fact that the reconstruction difficulty of different patches can vary sharply, and the semantic reasoning and local perception ability required for patches are not the same. Generally, recovering patches with more visible patches around will be simpler, as long as the model has sufficient local perception ability. In contrast, reconstruction with few visible patches around requires the model to have strong semantic reasoning ability, given that there is little access to neighboring information. Treating all pixels equally will neglect this demand for different properties of the model, inevitably limiting its representation ability. On the other hand, layers at different training phases naturally learn features at different levels. Therefore, we ask if there is a way to focus on objectives with diverse characteristics as training progresses so that better representations can be learned overall.

Motivated by this observation and answering the question above, we propose a simple yet effective SSL framework named **D**eep **D**ynamic **A**uto**E**ncoder (DDAE). With deep dynamic supervision, the model dynamically focuses on different patches at different training phases. First, it attaches importance to patches close to visible ones, which only demands fair local perception to recover. Then compared with the beginning of training, the model further pays attention to the distant and difficult patches aided by the simpler patches, which have been attached more importance in the earlier pretraining phase. Specifically, we first define the reconstruction difficulty according to the distance between masked patches and visible ones, generating a distance map. Then, we exert different supervision signals (which are controlled by learnable parameters $\beta$) for intermediate layers. As the training progresses, the model dynamically focuses on different regions with the update of $\beta$.

Since the pixel representations of perceptually similar images might be very different, only pixel-level reconstruction can be vulnerable to the perceptual difference between images (Dong et al., 2021). Besides pixel regression, we further apply feature self-distillation to provide semantic information. We directly align the encoder's intermediate features with the corresponding features of the momentum encoder in deep dynamic supervision, where the momentum encoder is updated by Exponential Moving Average (EMA) (Grill et al., 2020; He et al., 2020). It is worth noting that the feature distillation target here is the corresponding feature of the momentum encoder, while the regression targets for intermediate layers are all raw pixels. Dynamic loss is applied for both feature self-distillation and pixel regression.

Furthermore, we directly migrate the core design deep dynamic supervision to the representative methods in one-stage and two-stage, MAE (He et al., 2022) and BEiT-v2 (Peng et al., 2022a) respectively, surpassing original methods with nontrivial margins. Since deep dynamic supervision does not introduce any additional structure, it can be used as a general MIM pretraining plug-and-play module.

The contributions are summarized as follows:

- We propose DDAE, a one-stage tokenizer-free framework for self-supervised representation learning. Our core design deep dynamic supervision can be migrated to existing MIM approaches seamlessly, providing more local priors for ViTs to make up for their inherent structural defects.

- Although deep supervision does not perform well in modern supervised learning, DDAE combines dynamic loss and deep supervision in a novel and effective way, making a step towards unleashing the potential of deep supervision in MIM.

- The experimental results demonstrate the effectiveness of the proposed DDAE, we achieve state-of-the-art performance among tokenizer-free methods and have competitive performance with many two-stage methods.

## 2 RELATED WORK

**Local Prior for ViTs.** Aided by the global receptive field of multi-head self-attentions(MSAs) mechanism, ViTs (Dosovitskiy et al., 2020) have significantly advanced computer vision tasks such as image classification (Wu et al., 2020; Touvron et al., 2021a; Yuan et al., 2021), object detection (Carion et al., 2020; Zhu et al., 2020; Beal et al., 2020), and semantic segmentation (Zheng et al., 2021).However, since self-attention does not have CNN's local prior, it often ignores local feature details, which largely leads to ViT's demand for large dataset pretraining. To solve, DeiT (Touvron et al., 2021a) transfers CNN-based knowledge to vision transformer through a distillation token. T2T-ViT (Yuan et al., 2021) proposed using a tokenization module to recursively reorganize the image to consider neighboring pixels. Local MSAs (Yang et al., 2019; Liu et al., 2021; Chu et al., 2021) are proposed to calculate self-attention only within small windows, achieving better performance than global MSAs not only on small datasets but also on large datasets, *e.g.*, ImageNet-21K. Since local prior is the natural inductive bias of convolution, there are also works aimed at integrating convolution and vision transformer. Works like PVT (Wang et al., 2021)and CvT (Wu et al., 2021) insert spatial reduction or convolution before global self-attention, yielding the merits of self-attention and convolution. MixFormer (Chen et al., 2022a) and Conformer (Peng et al., 2021) integrate the advantages by building parallel interactions. It should be noted that all above methods fundamentally change ViTs' basic structure, bringing negative effects such as restricted receptive field or increased computational cost of downstream tasks. Different from them, our work focuses on providing local prior for vanilla ViTs without introducing any extra structural modification.

**Masked Image Modeling.** Inspired by BERT (Devlin et al., 2018) in NLP, MIM learn representation by reconstructing masked image patches from visible patches. Existing MIM methods can be divided into two categories according to the need for the additional tokenizer. **Two-stage methods:** Represented by the pioneering work BEiT (Bao et al., 2021), two-stage methods (Bao et al., 2021; Dong et al., 2021; Wei et al., 2022b; Chen et al., 2022b; Peng et al., 2022b) firstly transform image patches into semantic visual tokens through a pretrained discrete VAE (Ramesh et al., 2021)as visual tokenizer, then the corresponding tokens of masked image patches are reconstructed to pretrain the encoder. Nevertheless, the tokenizer needs to be offline pretrained with fixed model architectures and extra data (Zhang et al., 2019b; Ramesh et al., 2021; Radford et al., 2021), some methods even further require an off-the-shelf DNN as teacher to distill tokenizer pre-training (Peng et al., 2022b), which complicates the process and increases cost. **One-stage methods:** MAE (He et al., 2022)constructs an asymmetric encoder-decoder structure, and directly performs raw pixel regression of masked image patches. SimMIM (Xie et al., 2022b) allows hierarchical transformers such as Swin (Liu et al., 2021) to be directly applied to MIM. MaskFeat (Wei et al., 2022a), Data2vec (Baevski et al., 2022), BootMAE (Dong et al., 2022), sDAE (Chen et al., 2022c) explored the choice of reconstruction targets. However, existing MIM methods reconstruct all patches equivalently, which is obviously suboptimal. Several inpainting works explored the dynamic loss design (Pathak et al., 2016; Yeh et al., 2016; Yu et al., 2018; Wang et al., 2018), but there is essential difference between MIM and inpainting. Inpainting focuses on the quality of the generated image, while MIM focuses on the representation of the encoder obtained from pretraining. Our DDAE is a tokenizer-free framework that focuses on dynamic reconstruction at different pretraining phases.

**Deep Supervision.** Deep supervision methods are proposed to accelerate model's convergence and alleviate the problem of gradient vanishment. Through applying aux layers to transmit the supervision signal to the shallow layers, deep supervision has been used in early classification models (Szegedy et al., 2015; Lee et al., 2015) and extended to other visual recognition tasks (Xie & Tu, 2015; Newell et al., 2016; Zhang et al., 2018b; Mosinska et al., 2018; Zhang et al., 2018a). Despite these advances, modern CNN classification models rarely use auxiliary classifiers since directly appending simple auxiliary classifiers on top of the early layers of a modern network hurts its performance (Huang et al., 2017). Several works utilize deep supervision to enhance the network through knowledge distillation(Sun et al., 2019; Zhang et al., 2019a). Zhang et al. (2022) proposed Contrastive Deep Supervision to use contrastive learning signals for intermediate layers, but the signals of intermediate layers are still the same. Different from them, we introduce inconsistency in the

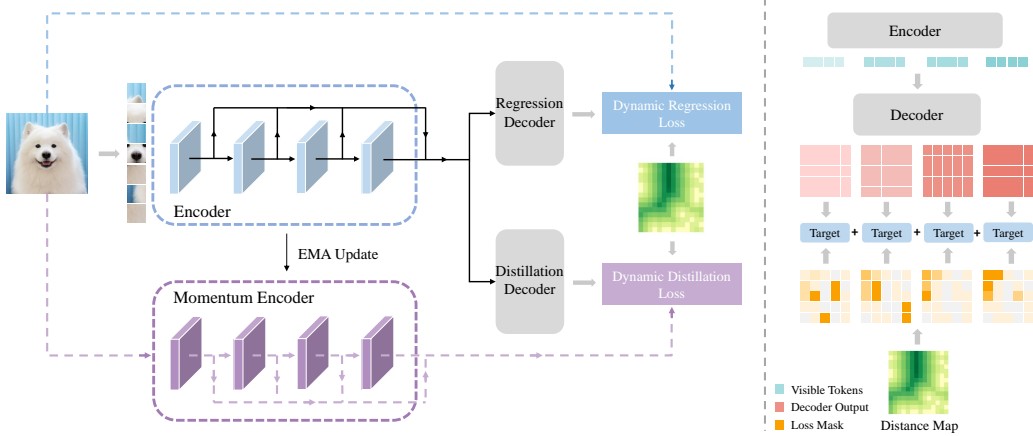

(a) Overall Framework of DDAE    (b) Deep Dynamic Loss

Figure 1: (a) Overall framework of DDAE. Only visible patches are fed into the encoder while full patches are fed into the momentum encoder. We perform raw pixel regression and feature self-distillation both with deep dynamic loss. (b)Deep dynamic loss. Lighter color indicates that the signal comes from the shallower layer of the encoder. Best viewed in color.

intermediate signals for the first time. To the best of our knowledge, this paper is also the first to apply deep supervision to the MIM task. State-of-the-art results show the great potential of deep supervision in the application of MIM.

## 3 METHODS

In this section, we firstly elaborate on the basic framework of MIM. Then we introduce our proposed DDAE's two core designs termed Deep Dynamic Supervision and Deep Self-Distillation in Section 3.2 and Section 3.3 respectively. Our framework is illustrated in Fig 1.

### 3.1 PRELIMINARY

Formally, MIM firstly divides the input image $X \in \mathbb{R}^{H \times W \times C}$ into non-overlapping flattened patches $\mathbf{x} = [\mathbf{x}_1, \mathbf{x}_2, ..., \mathbf{x}_N]$, where $\mathbf{x}_i \in \mathbb{R}^{P^2 C}$ according to the patch size $P$, and $N = (H \times W)/P^2$ is the number of patches. It then samples a random binary mask $\mathbf{m} = [m_1, m_2, ..., m_N]$, where $m_i \in \{0, 1\}$ to mask a portion of the flattened patches. The masked patches $\mathbf{x}_M \triangleq \mathbf{x} \odot \mathbf{m}$ are discarded (He et al., 2022) or substituted by a learnable token [MASK] (Bao et al., 2021), and the rest patches $\mathbf{x}_{1-M} \triangleq \mathbf{x} \odot (1 - \mathbf{m})$ are used to reconstruct the dropped features or images to learn rich representations. The optimization target of MIM can be formulated as follow:

$$\min_{\theta, \phi} \mathbb{E}_{X \sim D} \mathcal{M}(d_\phi([f_\theta(\mathbf{x}_{1-M}), [\text{MASK}] \odot \mathbf{m}]), \mathbf{x}_M) \qquad (1)$$

where $\odot$ denotes element-wise multiplication; $f_\theta(\cdot)$ and $d_\phi(\cdot)$ are encoder and decoder respectively; $\mathcal{M}(\cdot, \cdot)$ is the similarity measurement, which varies in different works, e.g., $l_2$-distance in pixel space (He et al., 2022), perceptual loss in codebook space (Dong et al., 2021) or self-distillation loss in feature space (Chen et al., 2022c). In our work, we use the $l_2$-distance as our measurement, $\mathcal{M}(a, b) = ||a - b||^2$, for both of the pixel reconstruction and self-distillation. To simplify the formulation, we ignore the mask token term and use $d_\phi(\cdot)$ to represent $d_\phi([\cdot, [\text{MASK}] \odot \mathbf{m}])$.

### 3.2 DEEP DYNAMIC SUPERVISION

**Dynamic.** Firstly, we define the difficulty of reconstruction according to the distance between each masked patch and visible patches, generating a *distance map* with a distance transform $D(\cdot)$. For

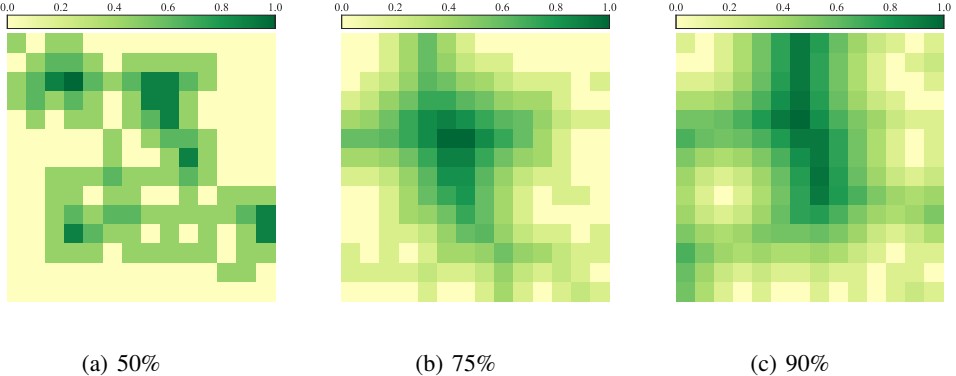

(a) 50%            (b) 75%            (c) 90%

Figure 2: Distance map (normalized to [0, 1]) under different mask ratio. Darker color means larger distance from visible patches. High mask ratio leads to diverse distance for masked patches.

each masked token ($m_i = 1$), the distance transform assigns a number that is the Euclidean distance between that token and the nearest unmasked token in 2D space. Naturally, it is difficult to recover a patch that is far from visible ones and demands stronger semantic reasoning ability. On the contrary, reconstruction of a patch with visible ones nearby only requires fair local perception.

As shown in Fig 2, existing MIM methods often use a high mask ratio like 75% (which is proved to be critical to MIM's success (He et al., 2022)), values of the distance map vary diversely. Since the distance map is based on patches ($14 \times 14$) rather than pixels ($224 \times 224$), it is cost-free.

To guide the model to focus on different patches (corresponding to the requirement of distinctive properties) in different training phases, we propose $h_\beta(\cdot)$ to learn a dynamic coefficient $\beta$ of the distance map to generate loss weight, directly applying dynamic focus to the loss weight of corresponding patches. $h_\beta$ is derived as follows:

$$h(m_i|\beta) = \begin{cases} \dfrac{\exp(D(m_i)^\beta)}{\sum_{m_j=1}\exp(D(m_j)^\beta)} & , m_i = 1 \\ 0 & , m_i = 0 \end{cases} \tag{2}$$

We re-scale the loss weight map into $[0, 1]$ according to the max value of the weights. When $\beta > 0$, larger distance leads to greater loss weight, so the model pays more attention to the reconstruction of difficult patches. Increasing the value of $\beta$ can exacerbate this trend. Conversely, when $\beta < 0$, larger distance leads to smaller loss weight, and decreasing $\beta$ results in more importance attached to simple patches. Along with the changing of $\beta$, the model dynamically focuses on patches with diverse degrees of recovery difficulty.

**Deep supervision.** As commented in the introduction, the reconstruction of patches with diverse difficulties requires distinct characteristics of the model, and layers at different depths naturally learn features at different levels. Therefore, we apply dynamic loss to the intermediate layers. Taking the standard ViT with $B$ blocks as the encoder, we divide the blocks into $K$ groups and extract the features at the end of each group. For example, we set $K = 4$, then we extract output features of block 3, 6, 9 and 12 and feed them into the decoder respectively to recover masked patches. $K$ is also called *supervisory number*, since $K$ is equal to the number of supervisory signals. Then the loss function of pixel reconstruction $L_{pixel}$ is derived as follows:

$$L_{pixel}(\theta, \phi, \beta) = \sum_{i=1}^{K} \lambda_i h_{\beta_i}(\mathbf{m}) \odot ||d_{\phi_p}(f_\theta^{(g_i)}(\mathbf{x}_{1-M})) - \mathbf{x}_M||^2 \tag{3}$$

where $f^{(i)}(\cdot)$ denotes to the output features of block-$i$ in encoder, and $g_i = \frac{B}{K}i$ denotes to the group index; $d_{\phi_p}$ is the regression decoder; $\boldsymbol{\lambda}$ are the scale factors of each group, and set to $[0.75, 0.75, 0.75, 1]$ by default. Note that $\beta$ of different layers are independent of each other.

Deep dynamic supervision does not introduce any additional structure, so it can be seamlessly migrated to existing MIM structures. We discussed its universality in Section 4.2 where we migrated it to the representative MIM methods of one-stage and multi-stage.

## 3.3 DEEP SELF-DISTILLATION

In addition to raw pixel regression, the layer-by-layer correspondence between intermediate features is more suitable for the design of deep dynamic supervision. Therefore, we designed Deep Self-Distillation based on BootMAE (Dong et al., 2022) to further strengthen the model through high-level semantic features. Specifically, the momentum encoder provides the features of masked patches of each layer as the target of self-distillation, so that the intermediate features of each layer corresponding to the encoder have their own targets of self-distillation. The momentum encoder is updated by the encoder using the exponential moving average method (EMA). Formally, denoting the parameters of the encoder $f_\theta$ as $\theta$ and those of the momentum encoder $f_{\theta'}$ as $\theta'$, we update $\theta'$ by:

$$\theta' \leftarrow m\theta' + (1-m)\theta \tag{4}$$

Here $m \in [0, 1)$ is a momentum coefficient and set to 0.9999 by default. Note that $\theta'$ are not updated by back-propagation but by equation 4 after each training step. The feature self-distillation also uses deep dynamic supervision as pixel regression. Note that the regression of raw pixels is one-vs-all, the features of all layers will be reconstructed by one regression decoder. The feature self-distillation is all-vs-all, which means the shallow features of the encoder are self-distilled by the corresponding shallow features of the momentum encoder, and the deep features are self-distilled by the corresponding deep features. Pixel regression and feature self-distillation use separate decoders, which both consist of two-layer transformer blocks. Since the decoders are light-weighted, the additional cost brought by deep supervision is acceptable. Now the deep self-distillation formula is:

$$L_{distill}(\theta, \beta) = \sum_{i=1}^{K} \lambda_i h_{\beta_i}(\mathbf{m}) \odot ||d_{\phi_d}(f_\theta^{(g_i)}(\mathbf{x}_{1-M})) - f_{\theta'}^{(g_i)}(\mathbf{x}_M)||^2 \tag{5}$$

where $f_\theta(\cdot)$ and $f_{\theta'}(\cdot)$ are encoder and momentum encoder respectively; $d_{\phi_d}$ is the distillation decoder; $\boldsymbol{\lambda}$ are the scale factors of each group, and set to $[0.75, 0.75, 0.75, 1]$ by default. We add an $L_2$ regularization term to constrain the magnitude of $\beta$. Since $L_2$ norm reaches minimum under uniform distributions, the $L_2$ regularization loss here is to ensure that the loss mask does not deviate too far from a uniform mask, this can effectively prevent beta from being too extreme(in which case the model only focus on the hardest or simplest patches).The overall loss function is:

$$L = L_{pixel} + L_{distill} + \lambda \sum_i ||h_{\beta_i}(\mathbf{m})||_2 \tag{6}$$

where $\lambda$ is the scale factor to tune the $L_2$ regularization term and is set to 0.5 by default.

## 4 EXPERIMENTS

### 4.1 IMPLEMENTATION

We conduct experiments on ImageNet-1K without labels as the pretraining data for self-supervised learning. The input resolution is set as 224×224 during pretraining and partitioned into 16×16 size patches. We pretrain the standard ViT small and base architectures, i.e., ViT-S/16 and ViT-B/16. The pixel regression decoder and feature distillation decoder both consist of 2 transformer blocks, along with an extra linear projection head for predictions. The dimension for the pixel regression decoder is 512 while for feature distillation decoder is set the same as the encoder. We use block-wise masking with a ratio of 75%. The data augmentation is only standard random cropping and horizontal flipping. All $\beta$ are initialized as $-0.5$ by default. The pretraining settings are almost the same as BootMAE (Dong et al., 2022) (See Appendix A for details).

### 4.2 IMAGE CLASSIFICATION

We evaluate both fine-tuning accuracy and linear probing accuracy on ImageNet-1k. Table 1 presents the comparison with previous state-of-the-art MIM-based methods including one-stage and two-stage. As a tokenizer-free framework, our DDAE acheives consistent advantages both in a short schedule and a longer schedule. In particular, with only 100 epochs pre-training, DDAE can achieve

Table 1: Image classification accuracy (%) comparison on ImageNet-1K of different methods with ViT-B as backbone. We report the fine-tuning and linear probing accuracy and our method DDAE outperforms previous self-supervised methods.

| Methods | Extra Tokenizer | Pretraining Epochs | Finetune | Linear |
|---|---|---|---|---|
| DINO (Caron et al., 2021) | contrastive | 300 | 82.8 | 78.2 |
| MoCo-v3 (Chen et al., 2021) | contrastive | 300 | 83.2 | 76.7 |
| SplitMask (El-Nouby et al., 2021) | w/ | 300 | 83.6 | N/A |
| CAE (Chen et al., 2022b) | w/ | 300 | 83.6 | 64.1 |
| MaskFeat (Wei et al., 2022a) | w/o | 300 | 83.6 | N/A |
| BootMAE (Dong et al., 2022) | w/o | 100 | 83.0 | N/A |
| SdAE (Chen et al., 2022c) | w/o | 100 | 83.5 | 60.3 |
| **DDAE(ours)** | w/o | 100 | **83.5** | 60.3 |
| BEiT (Bao et al., 2021) | w/ | 800 | 83.2 | 56.7 |
| MAE (He et al., 2022) | w/o | 800 | 83.4 | 64.4 |
| MAE (He et al., 2022) | w/o | 1600 | 83.6 | 68.0 |
| SimMIM (Xie et al., 2022b) | w/o | 800 | 83.8 | 56.7 |
| CAE (Chen et al., 2022b) | w/ | 1600 | 83.9 | 70.4 |
| MaskFeat (Wei et al., 2022a) | w/o | 1600 | 84.0 | N/A |
| BootMAE (Dong et al., 2022) | w/o | 800 | 84.2 | 66.1 |
| SdAE (Chen et al., 2022c) | w/o | 800 | 84.0 | N/A |
| **DDAE(ours)** | w/o | 800 | **84.3** | 67.7 |

comparable performance with MAE using 1600 epochs pre-training and surpass 800 epochs pretrained BEiT. Furthermore, with a longer pretraining schedule, DDAE achieves 84.3% top-1 accuracy, developing a new state-of-the-art on ImageNet-1K among one-stage self-supervised methods. Note that despite SdAE reaching comparable results with DDAE when pretrained for 100 epochs, our method outperforms it with + 0.3% gains with 800 epoch schedule, demonstrating our consistent advantages both in a short schedule and a longer schedule.

In terms of linear probing, our approach surpasses the above MIM methods with the same training epochs, but is not as good as the contrastive-based methods. Contrastive-based methods compare across images while MIM-based methods exploit the whole image structure, which may care about more than 1000 classes (Chen et al., 2022b). This phenomenon is also reported in MAE (He et al., 2022) and BEiT (Bao et al., 2021), so for MIM-based methods, fine-tuning measurement may be a better metric to validate their effectiveness.

## 4.3 ANALYSIS OF DDAE

**Migrating Deep Dynamic Supervision to Existing MIM methods.** Although for simplicity we build our framework as a one-stage approach, our core design, Deep Dynamic Supervision(DDS), is compatible with other existing MIM methods. To demonstrate that, we conduct experiments to migrate DDS into representative one-stage and two-stage MIM methods, MAE (He et al., 2022) and BEiT-v2 (Peng et al., 2022a) respectively. We extract intermediate features of the encoder, then feed them into decoder together with the encoder's final output. The reconstruction targets are all set the same as the final output. Reconstruction losses for intermediate layers are multiplied by their corresponding dynamic loss weight as mentioned in Section 3.2. The plug-and-play results reported in Table 2 demonstrates our design's generality for both one-stage and two-stage approaches.

**More local priors for ViTs.** We conjecture that the fine-grained task built by MIM enables ViT to pay more attention to local information besides using global semantic inference, which inherently makes up for the absence of local prior in the structure of ViTs. We depict the average attention dis-

Table 2: Plug-and-play Experiments.

| Methods | Extra tokenizer | Pretraining Epochs | Finetune |
|---|---|---|---|
| MAE (He et al., 2022) | w/o | 300 | 82.9 |
| DDS-MAE | w/o | 100 | **83.2** |
| BEiT-v2 (Peng et al., 2022a) | w/ | 100 | 83.9 |
| DDS-BEiT-v2 | w/ | 100 | **84.1** |

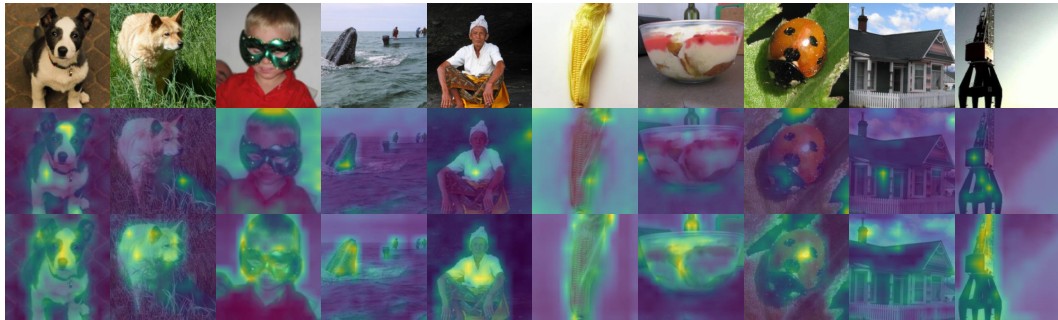

Figure 3: The attention map averaged over 12 attention heads between the class token and the patch tokens in the shallow layer (Block 3) of the ViT-B encoder pretrained on ImageNet-1K. Top: Input image, Middle: MAE, and Bottom: our DDAE. We can see that DDAE has formed effective target perception attention in the very early stage, while MAE still behaves to be rudimentary.

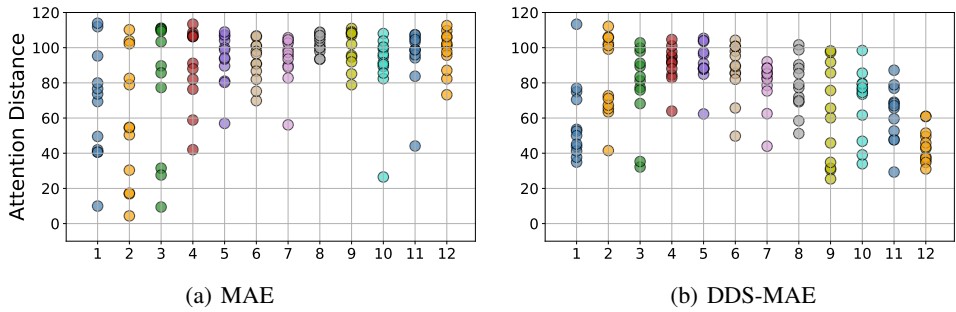

|           (a) MAE            |          (b) DDS-MAE          |

Figure 4: The averaged attention distance in different attention heads (dots) w.r.t the layer number on MAE (a) and DDS-MAE (b) with ViT-B as the backbone.

tance of vanilla MAE and DDS-MAE to demonstrate our design's effectiveness in providing more local priors in deep layers. Specifically, we compute averaged attention distance in each attention head of each layer (Dosovitskiy et al., 2020). We conduct experiments based on MAE since it is a simple enough MIM framework. As illustrated in Fig 4, DDS remarkably reduces the average attention distance of the model in deep layers. This phenomenon indicates that through deep dynamic supervision the model focuses more on local vicinity, thus obtaining more local priors in deep layers. Moreover, for DDS part, different heads in relatively deeper layers behave more diversely, which may also contribute to DDS's effectiveness (+ 1.22% gains over MAE on ViT-S and more than + 0.3% gains on ViT-B).

**Analysis on dynamic curve of $\beta$ and attention map.** We depict the curve of dynamic coefficients $\beta$ changes during training for different layers in Fig 5. For all intermediate layers, $\beta$ drops sharply at the beginning, then rises slowly as the training progresses. A smaller $\beta$ value means that the model pays more attention to simple patches which are close to visible ones. Therefore, in the overall trend, each layer attaches more importance to the reconstruction of simpler patches and gradually pays more attention to difficult ones. We further show the shallow layer's attention map of MAE and DDAE in Fig 3. Our DDAE forms target perception attention effectively in the very early stage, while MAE still behaves to be rudimentary.

**Ablation on Deep Dynamic Supervision.** We conduct ablation studies to analyze the contributions of each component in Deep Dynamic Supervision. The models are finetuned on ImageNet-1K for 100 epochs. We set ViT-S as encoder and pretrain for 100 epochs. Table 3 shows that either ablation degrades the performance, suggesting that both designs benefit the training dynamics.

Table 3: Ablation study on Deep Dynamic Supervision. Here D-MAE means applying Dynamic Loss to MAE, DS-MAE means applying Deep Supervision. ViT-S is used as backbone.

| Methods | Deep Supervision | Dynamic Loss | Finetune |
|---|---|---|---|
| MAE (He et al., 2022) | | | 78.7 |
| D-MAE | | ✓ | 78.9 |
| DS-MAE | ✓ | | 79.7 |
| DDS-MAE | ✓ | ✓ | **79.9** |

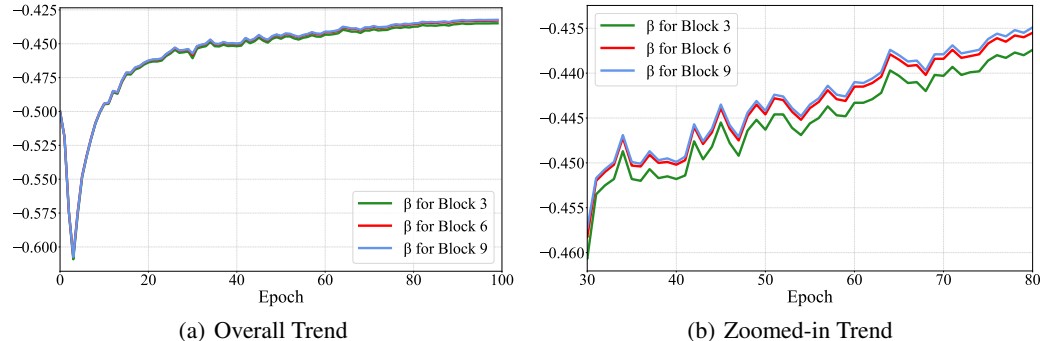

(a) Overall Trend          (b) Zoomed-in Trend

Figure 5: $\beta$ dynamic curve for intermediate layers. (a) is the overall trend from epoch 1 to epoch 100; (b) is the zoomed-in trend. Larger $\beta$ means more importance is attached to difficult patches.

Table 4: Ablation study on Deep Self-Distillation. Here ✓ means applying DDS on that part.

| Methods | Pixel Regression | Feature Self-Distillation | Finetune |
|---|---|---|---|
|  |  |  | 83.0 |
|  |  | ✓ | 83.2 |
|  | ✓ |  | 83.3 |
| DDAE | ✓ | ✓ | **83.5** |

**Ablation on Deep Self-Distillation.** Then, we study Deep Self-Distillation in DDAE, we conduct ablation experiments with ViT-B as encoder, pretrain for 100 epochs and finetune 100 epochs on ImageNet-1K. Results in Table 4 suggests that applying DDS on Pixel and Feature both can improve the feature representation learned.

## 4.4 SEMANTIC SEGMENTATION

We evaluate the learned representation of DDAE on the ADE20K benchmark (Zhou et al., 2019), which consists of 25K images and 150 semantic categories. We use the UperNet (Xiao et al., 2018) task layer for semantic segmentation. We train Upernet 160K iterations with single-scale inference. The results are reported in Table 5 with mean Intersection of Union (mIoU) as the evaluation metric. Compared with previous state-of-the-art self-supervised models, our proposed DDAE outperforms all the one-stage methods, further validating the effectiveness of our framework.

Table 5: Semantic segmentation mIoU (%) comparison on ADE20K. ViT-B is used as backbone.

| Methods | Extra Tokenizer | Pretraining Epochs | mIoU |
|---|---|---|---|
| BEiT (Bao et al., 2021) | w/ | 800 | 45.6 |
| SimMIM (Xie et al., 2022b) | w/o | 800 | 46.9 |
| MAE (He et al., 2022) | w/o | 1600 | 48.1 |
| PeCo (Dong et al., 2021) | w/ | 800 | 48.5 |
| CAE (Chen et al., 2022b) | w/ | 800 | 48.8 |
| SdAE (Chen et al., 2022c) | w/o | 800 | 49.0 |
| BootMAE (Dong et al., 2022) | w/o | 800 | 49.1 |
| **DDAE(ours)** | w/o | 800 | **49.3** |

## 5 CONCLUSION AND LIMITIATIONS

In this paper, we propose a new framework DDAE with two core designs. (1) We present deep dynamic supervision to enable dynamic focus on different patches through pretraining, providing progressively richer representation. (2) We propose to self-distill intermediate features of momentum encoder with dynamic supervision, which enhance the representation from a high semantic level. More importantly, we find our approach can bring more local priors for ViTs in deep layers, which we empirically find crucial for MIM's success. Although our DDAE is effective, the correlation between difficulty and distance is less relevant for background patches. How to effectively distinguish foreground and background in an unsupervised way and apply different weights is a problem that needs to be further explored.

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

## A    TRAINING DETAILS

**ImageNet Pretraining.** We use Adam (Kingma & Ba, 2014) as the optimizer and train the DDAE for 100/800 epochs to illustrate both the rapid convergence and a higher upper limit. The learning rate is set as $1.5e^{-4} * batchsize/256$ with cosine learning rate decay and warmed up for 20/40 epochs respectively. The total batch size is set as 4096. The weight decay is set as 0.05. We increase the momentum parameter from 0.999 to 0.99999 in the first 400 epochs. The mask ratio is 75%.

**ImageNet Fine-tuning.** We follow the fine-tuning protocol in BEiT to use layer-wise learning rate decay as 0.75/0.65 for ViT-S/ViT-B respectively. The batch size is 1024, the warmup epoch is 5 and the weight decay is 0.05. For ViT-S, we train 100 epochs with a learning rate 1.6e-2. For ViT-B, We train 100 epochs with a learning rate 8e-3.

**Linear probing.** We use the LARS (Karpathy et al., 2014) optimizer with momentum 0.9. We train the model for 90 epochs with a batch size of 16384. The warmup epoch is 10 and the base learning rate is 0.1. We set weight decay as zero.

**ADE20K Semantic segmentation.** We use UperNet (Xiao et al., 2018) based on mmsegmentation (Contributors, 2020). Following BEiT, we use AdamW (Loshchilov & Hutter, 2017) optimizer with initial learning rate $4e^{-4}$ , weight decay of 0.05 and batch size of 16 for $160K$ iterations. We warm up the learning rate for 1500 iterations and use a linear decay strategy. We set the layer decay as 0.65. Since ViTs can not output hierarchical features, we utilize four different scales FPNs to scale the feature map into different sizes. The input resolution is $512 \times 512$. We report single-scale test results.

## B    MORE ABLATION STUDIES

**Ablation on the supervisory number $K$.** We further explored the impact of supervisory number $K$ on performance. Specifically, we set $K$ to be 1, 2, 4, and 6 respectively. The method to determine the block index is the same as that described in Section 3.2. As shown in Table 6, the additional benefits brought by more than four supervisory signals have been saturated. So we set $K = 4$ as default in our framework.

Table 6: Ablation study on supervisory numbers $K$. Note all supervisions use Dynamic Loss.

| $K$ | **Block Index** | **Finetune** |
|---|---|---|
| 1 | $[12]$ | 78.9 |
| 2 | $[6, 12]$ | 79.5 |
| 4 | $[3, 6, 9, 12]$ | **79.9** |
| 6 | $[2, 4, 6, 8, 10, 12]$ | **79.9** |

**Ablation on the choice of $\beta$.** We also explored the choice of $\beta$. Firstly, we fix $\beta$ with different initialization, then we set $\beta$ as learnable parameters. Table 7 demonstrates that learnable $\beta$ benefits most. We also draw the $\beta$ curves with different initialization in Fig 6, the final values of $\beta$ and the relative relationship among block 3,6,9 tend to be stable and robust to initialization. We take $\beta = [-0.5, -0.5, -0.5, -0.5]$ as initialization in our experiments by default.

Table 7: Ablation study on the choice of $\beta$. Fixed means $\beta$ will not be updated in the pretraining.

| $\beta$ | **Initialization** | **Finetune** |
|---|---|---|
| Fixed | $[0, 0, 0, 0]$ | 79.7 |
| Fixed | $[-0.5, -0.5, -0.5, -0.5]$ | 79.5 |
| Learnable | $[0, 0, 0, 0]$ | 79.6 |
| Learnable | $[-0.1, -0.3, -0.5, -0.7]$ | 79.5 |
| Learnable | $[-0.7, -0.5, -0.3, -0.1]$ | 79.7 |
| Learnable | $[-0.5, -0.5, -0.5, -0.5]$ | **79.9** |

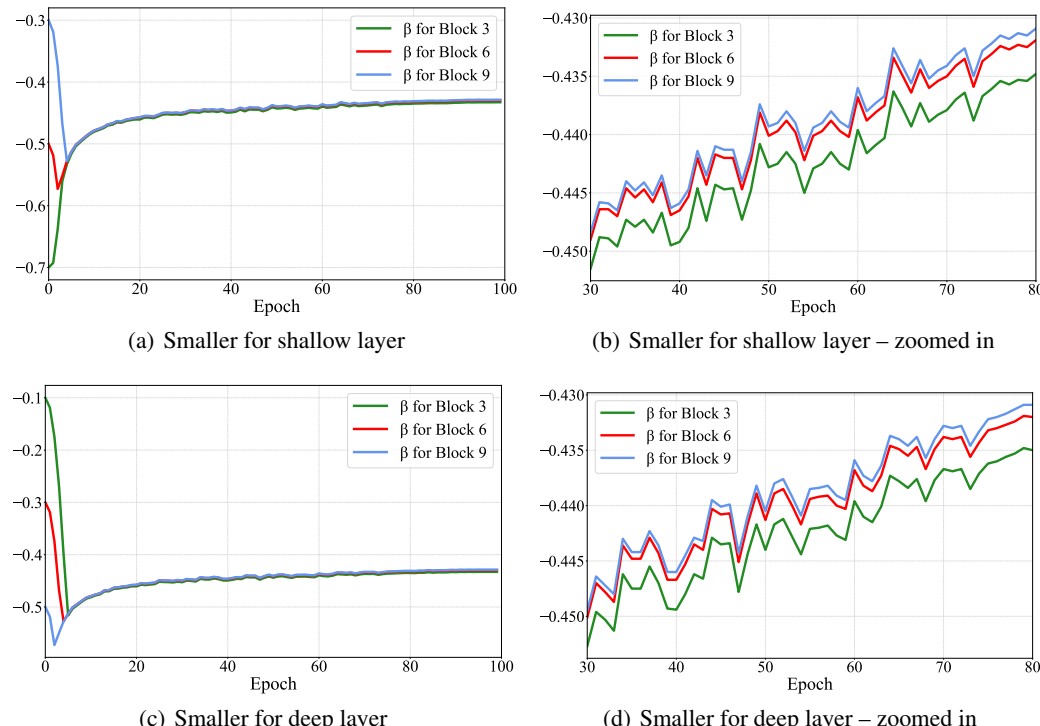

(a) Smaller for shallow layer

(b) Smaller for shallow layer – zoomed in

(c) Smaller for deep layer

(d) Smaller for deep layer – zoomed in

Figure 6: $\beta$ dynamic curve for different initialization. (a) Smaller values for shallow layer; (b) Smaller values for shallow layer – zoomed in; (c) Smaller values for deep layer; (d) Smaller values for deep layer – zoomed in. Different initialization ends with similar values of $\beta$ and relative relationship among different blocks.

## C  MORE VISUALIZATIONS

Our DDAE can form target-aware attention at a very early stage (Block 3), which is in sharp contrast to MAE. Here we show more shallow attention maps of DDAE and MAE in Fig. 7.

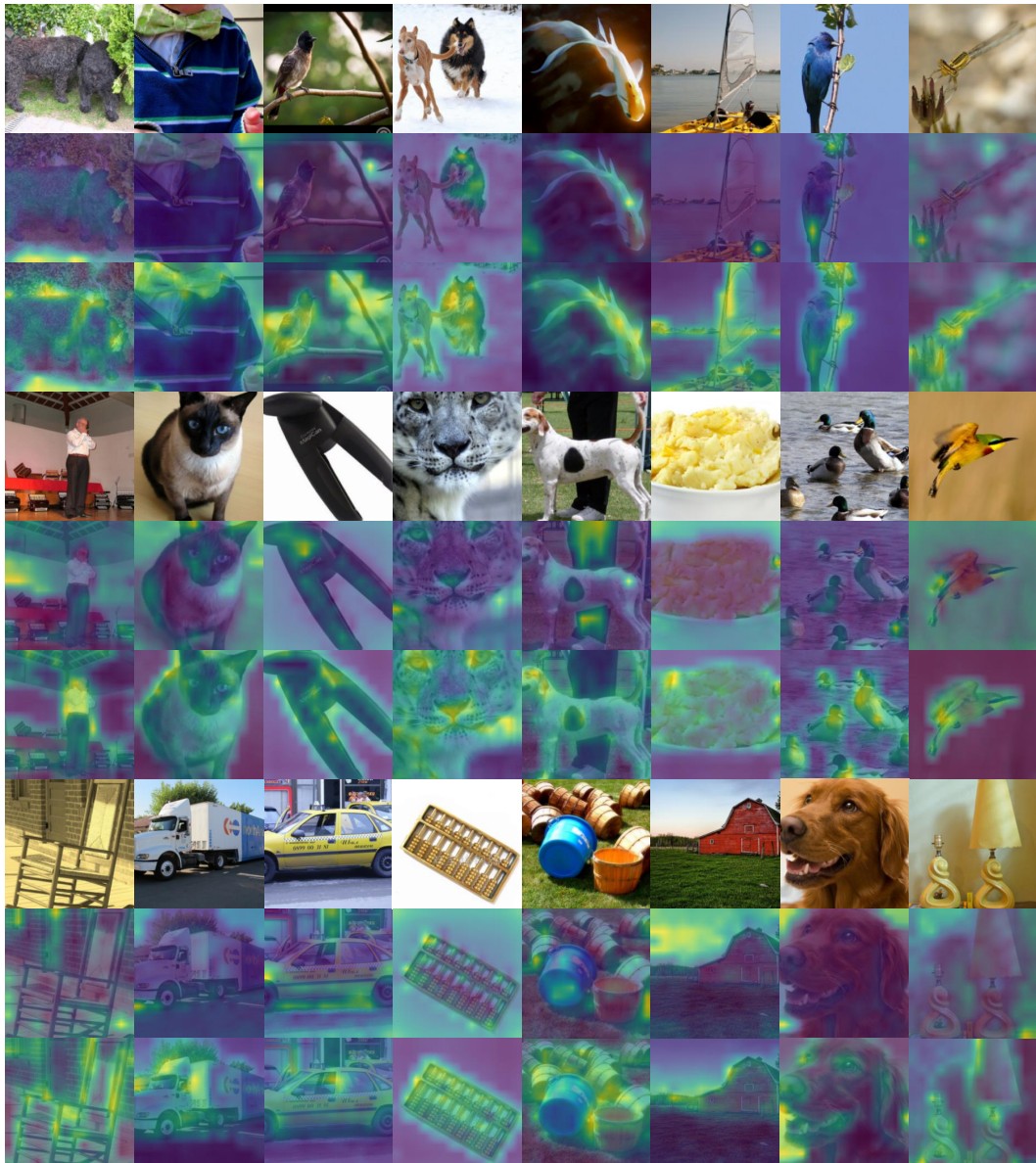

Figure 7: More attention map in the shallow layer (Block 3) of the ViT-B encoder of DDAE and MAE. The attention map is averaged over 12 attention heads between the class token and the patch tokens of the ViT-B encoder pretrained on ImageNet-1K. Top: Input image, Middle: MAE, and Bottom: our DDAE.

