# OpenReview forum: "Deep Dynamic AutoEncoder for Vision BERT Pretraining"
_ICLR.cc/2023/Conference — Submitted to ICLR 2023_

### Official Review · Reviewer_A9rs · 2022-10-22

**Confidence:** 4
**Correctness:** 4
**Technical Novelty And Significance:** 3
**Empirical Novelty And Significance:** 3
**Recommendation:** 5

**Clarity, Quality, Novelty And Reproducibility:**

The paper is well-written, and easy to follow. I suppose the analysis of deep supervision could be improved for a better understanding about the role of the momentum encoder.

About the novelty, as described in Strength And Weaknesses, the dynamic loss and momentum encoder are intensively studied in the inpainting and contrastive learning areas respectively. Their introduction and combination to MAE lead to interesting performance improvement with simple designs, which is inspiring.

The method description is clear and concise, and the related techniques are widely implemented in different topics. Thus, I believe this work is easily reproducible.

**Strength And Weaknesses:**

Strengths:
1) It gives a strong motivation for the proposed two designs. Applying spatially variant reconstruction losses to different masked patches based on their distances from the existing ones is a sound technique. The momentum encoder is also a typical method to improve representation learning.
2) It conducts extensive and solid experiments. Classification and semantic segmentation are conducted along with several compared methods. Ablations are also sufficient.

Weaknesses:
1) Though dynamic loss is complementary to deep supervision (in Table 3), the sole improvement brought by dynamic loss is limited (only 0.2 compared to MAE) and the major benefits are provided by deep supervision. This experimental result weakens the key role of dynamic loss while the loss is stressed with the most pages. Undoubtedly, dynamic loss can accelerate training convergence, but it is expected to give better improvement if it is the most crucial part of this paper. Besides, such a dynamic loss design is a routine in inpainting literatures [1-4].
> [1] Pathak, Deepak, et al. Context encoders: Feature learning by inpainting. CVPR. 2016.
>
> [2] Yeh, Raymond, et al. Semantic image inpainting with perceptual and contextual losses. arXiv preprint arXiv:1607.07539 2.3. 2016.
>
> [3] Yu, Jiahui, et al. Generative image inpainting with contextual attention. CVPR. 2018.
>
> [4] Wang, Yi, et al. Image inpainting via generative multi-column convolutional neural networks. NeurIPS. 2018.

2) The analysis of the used momentum encoder is unsatisfying from my perspective. If we treat the masked input as a view of the input, then MAE can be considered as maximizing the similarity between the positive pair in contrastive learning. To this end, the usage of momentum encoder can be discussed in depth.

**Summary Of The Paper:**

This paper proposes two practical regularization techniques for MAE training and the related quantitative results verify their efficacy. One is to exert different weights to the reconstruction loss of different patches based on how difficult it is to reconstruct these masked ones with visible ones. Another is to employ a momentum encoder to regularize the learned intermediate features. Experiments show these two designs are effective and complementary. They can accelerate training and further improve baseline performance, evaluated on two tasks, e.g., classification and segmentation.

**Summary Of The Review:**

This paper gives two simple designs, a dynamic loss and deep supervision, to improve the current MAE training and downstream performance. These two designs are widely researched in inpainting and contrastive learning topics, and they perform quite well on MAE, especially on classification and semantic segmentation. The motivations are well-described and well-supported by the experiments. I believe it gives good practices for the development of MAE.

---

> ### Author Response · Authors · 2022-11-08
> **Response to Reviewer A9rs**
>
> We sincerely thank the reviewer for the constructive comments. We have revised the paper accordingly (please see the revised version) and provided our detailed response below.
>
> **Weakness 1：**
> 	We give a more detailed description of dynamic loss, because in this paper, deep supervision is a simpler and more direct concept, which is easier to understand. But the generation of dynamic mask is more complex than deep supervision, and it is closely combined with deep supervision in this paper. If we do not clearly discuss dynamic loss, it will cause difficulty and vagueness in understanding our framework. We agree that the improvement brought by deep supervision is greater, but dynamic loss is also beneficial to MIM mechanism and more importantly, they are complementary to each other. Distinctive from existing MIM methods, Our work not only opened the door to explore dynamic MIM mechanisms, but also interestingly found that the combination of dynamic mechanisms and deep supervision can effectively enhance MIM, which can inspire the further research.
>
> Thank you for the four related articles on dynamic loss, we can learn from related dynamic loss designs to improve our framework and we have cited them in the related work part 2. Though dynamic loss design is a routine in inpainting literatures, in MIM it is not explored and there are several essential differences between MIM and inpainting. For example, inpainting focuses on the quality of the generated image, while MIM focuses on the representation of the encoder obtained from pretraining. Inpainting aims at restoring the original image as much as possible, while for MIM a multimodal distribution solution space may be more conducive to learning a general representation (for example, the pixel representations of perceptually similar images might be very different). On the other hand, the connection between inpainting and MIM is very close since they both do generative task from visible image, so we can learn from inpainting in some aspects like the dynamic loss in the future work. Thanks again for your valuable suggestions.
>
> **Weakness 2：**
> 	In fact, there has been relevant works [1, 2] trying to discuss the connection and unity of MIM and contrastive learning. The input of Momentum encoder is the full image, and all layers can output $X_M$. This characteristic naturally provides self-distillation targets for the encoder. Since the decoder generates $X^{'}_ {M}$ from$ X_{1-M}$, its output $X^{'}_ {M}$ constitutes positive pairs with the momentum encoder’s output $ X_M$. Moreover, since the features between the momentum encoder and the encoder are corresponding layer by layer, it is naturally suitable for deep supervision.
>
> As for this work, **we focus on building a dynamic reconstruction mechanism for MIM**, not only for feature self-distillation with the momentum encoder, but also suitable for regression of raw pixels and prediction of semantic tokens. In this work the momentum encoder plays a role in providing targets for deep feature self-distillation, similar as recent work [3] and [4]. We demonstrate the effectiveness of deep supervision on momentum encoder in ablations as shown in Table 4. As a crucial part of contrastive learning, momentum encoder also plays an important role in MIM, which is closely related to the integration of contrastive learning and MIM into a unified framework.
>
> [1] How mask matters: Towards Theoretical Understanding of MAE. NeurIPS 2022.
> [2] Understanding Masked Image Modeling via Learning Occlusion Invariant Feature. arXiv preprint arXiv:2208.04164, 2022.
> [3] SdAE: Self-distillated Masked Autoencoder. ECCV 2022.
> [4] Bootstrapped Masked Autoencoders for Vision BERT Pretraining. ECCV 2022.
>
> Thank you for summarizing that our work gives good practice for the development of MAE, and we sincerely hope that you can raise your score to be postive (since 5 is negative in ICLR). Please do not hesitate to let us know if you have any additional clarifications or experiments that we can offer, as we would love to convince you of the merits of our work.
>
> Sincerely,
>
> Authors

---

> ### Author Response · Authors · 2022-11-16
> **Looking Forward to Your Reply**
>
> Dear Reviewer A9rs：
>
> Thank you again for the suggestions, which have greatly helped us improve the paper.
>
> We have revised the paper as suggested (please see the revised version) and responsed to your concerns and questions as the response below, but have not heard your post-rebuttal responses yet.
>
> Please do not hesitate to let us know if you have any additional clarifications or experiments that we can offer, as we hope to have an effective discussion with you to address your concerns and to convince you of the merits of our work.
>
> Sincerely, Authors

---

### Official Review · Reviewer_VeHD · 2022-10-27

**Confidence:** 5
**Clarity, Quality, Novelty And Reproducibility:** The paper is easy to read. The experi…
**Correctness:** 3
**Technical Novelty And Significance:** 2
**Empirical Novelty And Significance:** 2
**Recommendation:** 3

**Strength And Weaknesses:**

Strength
+ The performance is good
+ Detailed experiments demonstrate the effectiveness of each component

Weakness
+ The idea seems to be incremental. Add deep supervision is straight-forward and should be expected to work.
+ It lacks motivation on using both of distillation decoder and regression decoder. It seems minor improvement when combing them. What is the experiment variance?
+ Using extra tokenizer is not necessary a bad thing. Will proposed method be further improved when use extra tokenizer?
+ What is the improvement on down-stream fine-tuning when compared to MAE?

**Summary Of The Paper:**

This paper designs a deep dynamic supervision mechanism that can be migrated into existing MIM methods.
It proposes to dynamically focus on patch reconstructions with different degrees of difficulty at different pretraining phases and depths of the model.
Further experiments demonstrate the effectiveness of proposed method.

**Summary Of The Review:**

Overall, this paper is valid. However, the idea seems to be incremental. This prevents me from further increasing the score.

After reading the rebuttal, I still feel that this paper lacks enough novelty and the main components proposed yeild minor improvements.

Hence, I downgrade my rating and would not recommend to accept.

---

> ### Author Response · Authors · 2022-11-08
> **Response to Reviewer VeHD**
>
> We sincerely thank the reviewer for the constructive comments. We have revised the paper accordingly (please see the revised version) and provided our detailed response below.
>
> **Weakness 1：**
> 	As introduced in related work part 3, **deep supervision does not work well in modern supervised learning** since directly appending simple auxiliary classifiers on top of the early layers of a modern network hurts its performance [1]. In this sense, whether deep supervision will work for MIM is not that straight-forward and our work introduces it into MIM in a novel form of deep dynamic supervision, successfully verifying its effectiveness in downstream tasks. **More importantly**, our contribution is not limited to the first introduction of deep supervision into MIM, **we propose a dynamic mechanism that enables MIM to focus on different patches’ reconstruction dynamically to learn better representation, which has been neglected by all the current approaches.** Our discussion and framework on this problem have opened the door to explore a more reasonable and dynamic pretraining mechanism for MIM. DDAE combines dynamic loss and deep supervision in an interesting and effective way, which is supported by the experimental results and appreciated by reviewer pPUj(makes a lot sense),reviewer x4N6(novel) and reviewer A9rs (gives a strong motivation).
>
> [1] Multi-scale dense networks for resource efficient image classification. ICLR 2018.
>
> **Weakness 2：**
> 	Thank you for raising this valuable question, we would like to clarify that +0.3% top-1 accuracy gains on ImageNet without extra tokenizer is not marginal. For instance, CVPR 2022 paper MAE [1] outperforms ICLR 2022 paper BEiT [2] by +0.2% gains with 800epoch pretraining, ECCV 2022 paper SdAE [3] surpasses MAE with +0.4% gains with 800 epoch pretraining.
>
> Since the pixel representations of perceptually similar images might be very different, only pixel level reconstruction can be vulnerable to the perceptual difference between images. So we further apply feature self-distillation to provide more semantic information. We keep the pixel regression branch to provide guidance for the model to reason about low-level textures. Moreover, the pixel regression also serves as a regularization to prevent collapse (in which case the encoder and the momentum encoder simply outputs constant).Both of them show effective performance improvement（+0.2% and +0.3%），and combining them leads to consistent and non-trivial improvement(+0.5%).
>
> As for the experiment variance, it is a common practice to run experiments only once on ImageNet because 1) ImageNet has 1281k images and training on such a large-scale dataset usually yields stable results; 2) running such experiments multiple times would be too expensive and environmentally unfriendly. For example, there is no experiment variance reported in BEiT [2], MAE  [1] and SdAE [3].
>
> [1] Masked Autoencoders Are Scalable Vision Learners. CVPR 2022.
> [2] BEiT: BERT Pre-Training of Image Transformers. ICLR 2022.
> [3] SdAE: Self-distillated Masked Autoencoder. ECCV 2022.
>
> **Weakness 3：**
> 	We agree and the answer for further improvement with extra tokenizer is yes. Due to computational resource limits and for simplicity, we build our framework as a tokenizer-free method, but we show that deep dynamic supervision can be migrated into existing two-stage method with strong extra tokenizer in Table 2. This demonstrates the complementarity of our approach to bring additional improvements over strong extra tokenizer.
>
> **Weakness 4：**
> 	Our DDAE outperforms MAE with a significant margin in downstream finetuning. For classification on Imagenet, **100** epochs pretrained DDAE achieves 83.5% top-1 accuracy, surpassing **800** epochs pretrained MAE (83.4%); 800 epochs pretrained DDAE gets **84.3%**, surpassing 1600 epochs pretrained MAE (83.6%) by a large margin (**+0.7% gains**). For semantic segmentation on ADE20K, DDAE achieves 49.3% mIoU for 800 epochs pretraining, surpassing MAE’s 48.1% mIoU with 1600 epoch pretraining significantly (**+1.2% gains**). The detailed contrast is in Table 1 and 5.
>
> Sincerely,
>
> Authors

---

### Official Review · Reviewer_x4N6 · 2022-10-28

**Confidence:** 4
**Clarity, Quality, Novelty And Reproducibility:** They have been discussed in the last …
**Correctness:** 2
**Technical Novelty And Significance:** 3
**Empirical Novelty And Significance:** 2
**Recommendation:** 6

**Strength And Weaknesses:**

Strengths:
+ I am familiar with and like deep supervision and distance transforms. They are techniques specific to computer vision due to their links to hierarchical representations and spatial affinity. This idea (if better developed) is something that I would like to cite and follow. Introducing deep supervision into masked image modeling is novel to my knowledge. Using distance transforms to weight training is a novel add-on too and makes sense in this masked image modelling setting.
+ Although the method is unclear in several places, I can somewhat figure out how to implement it as the modules are standard ones.
+ Although lacking in significance, the experimental results are conclusively better than baseline.
+ The attempt to reveal the underlying mechanism is appreciated (fig.3 and 4), although I am not fully convinced.

Weaknesses:
- I think the major issue is clarity:
- The notations make me confused. In equation 1, we use x_{1-M} to reconstruct x_M. But in equation 2 and 3, it seems that the paradigm changes? For equation 2, I believe we should reconstruct masked-out patches x_M right? Correct me if I am wrong. For equation 3, I think it is possible to distill the features of masked-out patches or remaining patches. But since the equation does not contain decoder I guess the input should be remaining patches x_{1-M} ? Having that said, if the distillation is done with only the encoder, how is the 'distillation decoder' in Fig.1 used?
- Reading section 3.3 makes me lost. There is no formal description of the EMA procedure or the training loop. And what's worse, the figure, the loss and the notation seem to contradict each other. The sentence 'Deep dynamic supervision is shared with raw pixel regression' brings much more confusion. What is 'shared with'? According to Fig.1 I guess the deep dynamic supervision term refers to the whole training objective?
- I think the beta regularization is the key to the success. According to my own experience, training this kind of parameters is not easy. The impact of this regularization needs more (theoretical and experimental) analysis. Since beta can be either negative or positive, regularizing its L1-norm is not valid. So I can see the authors use h for regularization, but the behavior is not readily clear to me.

- Although I think the method is novel from several points of view, I have to point out several missing references. In the 'Deep Supervision' related works section, the authors claim that making deep supervision work for modern architecture is not easy. I agree and I would like to point out that papers like [A][B], which are the referred 'Contrastive Deep Supervision' paper's baselines, have studied the problem.

[A] Deeply-supervised knowledge synergy, CVPR 2019
[B] Be your own teacher: Improve the performance of convolutional neural networks via self distillation, ICCV 2019

- My last major concern is about significance:
- Personally, I only care about whether these pre-trained models can help my group develop better down-streaming robotic scene understanding algorithms. I feel excited about CLIP, Swin or MDETR, but the performance improvement of this one seems too marginal. I don't think this is a strong enough pre-training paper for ICLR. Maybe other colleagues with hands-on expertise on large-scale pre-training can better assess the significance.

I have other comments:
- I doubt he central claim of 'locality inductive bias in deeper layers'. I am not convinced why we we should have local bias in late stages. Meanwhile, I cannot understand why the deep dynamic supervision design allows this to happen.
- 'gradually reconstructed vicinity' I am not convinced. There is no technical modules that allow gradual reconstruction.
- 'this paper shows another potential use of deep supervision to make the network more diverse' This is too vague and I am not convinced by this 'diverse' claim.
- The online calculation of distance transforms of random masks brings computation overhead, which should be analyzed.
- The better convergence claim should be supported by curves instead of two datapoints.

Several capical issues:
- The Experimental results
- Dynamic Loss
- The Reconstruction targets

**Summary Of The Paper:**

This manuscript proposes to introduce deep supervision into masked image modelling. Technically, there are two loss functions: the first one is to reconstruct pixel densities using tokens produced by intermediate transformer encoders; the second one seems to be enforcing the transformer encoder outputs to mimick those of corresponding outputs of an EMA version of the encoder (although I am not very sure about the details due to clarity issues). The primary selling point is to use a transformed (by equation 2) distance map that weights both losses so that patches with different distances to visible ones get different weights. This technique is considered 'dynamic' by the authors, in the tile, as the concentration parameter beta is learned in a layer-wise manner. Evaluations are done with downstreaming ImageNet fine-tuning and linear classification and an ADE20k fine-tuning experiments, reporting marginally improved performance. Codes are not provided or promised.

[Post-rebuttal]
The method is conceptually novel and has some value.
The significance is left for experts on large-scale pre-training to judge.
The major issue is unclear presentation. Although I can guess how to implement it from these vague descriptions, people who are not familiar with topics involved may have some difficulty in understanding this. This is sub-bar for an ICLR paper, in terms of clarity.
After several rounds of communication, the authors correct some factual errors (1-M v.s. M), and typos. They also tune-down some unjustified interpretations that may be misleading for the community. Remaining issues include:

(1) The behavior of the regularization term in Equation.6 lacks clarity. The authors give an intuitive example in the response, which is not included into the paper. If included, this raises the clarity of this part above the bar. But I still recommend an analysis, as mentioned before: 'The impact of this regularization needs more (theoretical and experimental) analysis' in the first round and 'How gradients propagate to beta, through h() is not clear to me and this needs an analysis' in the second round. This is not a difficult thing, which only needs some gradient analysis and I don't understand why authors insist it is not necessary.
(2) The training loop. After some clarification, I know that the implementation is aligned with my guessing. But the paper still lacks clarity. Including a training loop (like Algorithm 1 in the MoCo paper) is not difficult and I dont' understand why authors insist it is not necessary.

Although it is disappointing that the authors still ignore my sincere suggestions on improving the clarity for a larger audience that are not familiar with these techniques, I have raised the recommendation to BA.

**Summary Of The Review:**

The idea is fine, but the current version lacks in clarity. As for significance, I am not fascinated and others with hands-on experience in pre-training may have a better judgement.

---

> ### Author Response · Authors · 2022-11-08
> **Response to Reviewer x4N6**
>
> We sincerely thank the reviewer for the constructive comments. We have revised the paper accordingly (please see the revised version) and provided our detailed response below.
>
> **Weakness-clarity：**
>   	We are sorry for the confusion. There are some mistakes in the article due to hurry and negligence. In equation 3 and 4, $X_{1-M}$ and $X_{M}$ are reversed. In equation 4, the distillation is done with the distillation decoder, which we missed in the formula. We corrected these problems in the revised version(the modifications are highlighted with blue color). The sentence ‘Deep dynamic supervision is shared with raw pixel regression ’means feature self-distillation also uses deep dynamic supervision as pixel regression (i.e., deep dynamic supervision is utilized for both regression loss and distillation loss). We have revised this sentence, please see the revised paper.
>
> **Weakness-regularization：**
> 	We would like to clarify that we use L2-norm for regularization, so it can work with beta being positive or negative. Although both$ ||x|| $and $||x||_2$ can represent L2-norm, we modify equation 5 to the latter one for clarity. Here h is the function generating dynamic loss mask from distance map as defined in equation 2.
>
> **Weakness-Reference：**
> 	Thanks for your suggestion. We think these two articles are very valuable. They have been cited in related work as references in the revised paper.
>
> **Concern-significance：**
> 	We agree that the empirical improvement is not as impressive as CLIP or BERT. But we would like to clarify that +0.3% top-1 accuracy gains over ECCV 2022 SOTA SdAE [1] on ImageNet without extra tokenizer is not that marginal. For instance, SdAE surpasses CVPR 2022 paper MAE with +0.4% gains with 800 epoch pretraining.
> Due to computational resource limits and for simplicity, we build our framework as a tokenizer-free method, but we show that deep dynamic supervision can be migrated into existing two-stage method with strong extra tokenizer in Table 2. This demonstrates the complementarity of our approach to bring additional improvements over strong existing frameworks. Moreover, our core contribution is to point out an important problem ignored by the current MIM framework: reconstructing all patches equivalently. We propose deep dynamic supervision to solve this problem and have achieved effective results. Our discussion and framework on this problem opened the door to explore a more reasonable and dynamic pretraining mechanism for MIM.
>
> [1] SdAE: Self-distillated Masked Autoencoder. ECCV 2022.
> [2] Masked Autoencoders Are Scalable Vision Learners. CVPR 2022.
>
> **Concern-locality for deep layers：**
> 	In the finetuning, the model naturally learns local information at shallow layers and semantic information at deep layers. Therefore, the deep layer of ViT is usually lack of locality inductive bias. Since local prior is of vital importance for ViTs as introduced in the related work part 1, it is important to build a **complementary pre-training mechanism** to inject local prior into ViTs’ deep layers. Our method empirically makes up for the absence of local prior for ViTs through pretraining, thus demonstrating effective improvements on downstream finetuning tasks like classification and semantic segmentation. As for why, we conjecture that deep dynamic supervision enables model to focus on simpler patches first, then for difficult patches, their vicinity (i.e., simpler patches) has been reconstructed more or less, so the reconstruction of difficult patches can utilize neighboring information, does not rely solely on distant visible patches. In this way, the model can get more local prior in deep layers.
>
> **Concern-diversification：**
> 	We agree that the claim for diversification is not strongly supported by the experiments. Since dropping this claim does not diminish our contributions for proposing dynamic reconstruction mechanism in MIM and migrating deep supervision in an effective way, we do not claim for diversification anymore in the revised version to make the paper more rigorous.
>
> **Concern-computation overhead：**
> 	As introduced in Section 3.1, MIM firstly divides the input images into non-overlapping flattened patches. Distance map in our method is based on patches(14×14) rather than pixels(224×224), thus the computational overhead brought by calculating 14x14 map’s Euclidean distance is negligible.
>
> **Concern-convergence：**
> 	Thanks for your advice, we report 100 epoch and 800 epoch top-1 accuracy on ImageNet to showcase DDAE’s fast convergence and higher upper limit, which is representative for a fast schedule and a longer schedule in MIM. Considering the computational cost, we will draw a more detailed curve with more datapoints in the future version. So in the revised version, we replace the word ‘fast convergence’ to avoid ambiguity, the modified sentence is ’It reflects our consistent advantages both in a short schedule and a longer schedule.’
>
> Sincerely,
>
> Authors

---

> > ### Comment · Reviewer_x4N6 · 2022-11-14
> > **Response**
> >
> > (1) Correcting the masked/unmasked notation is appreciated.
> >
> > (2) Rephrasing this unclear 'shared' sentence is appreciated.
> >
> > (3) It is not about L1 or L2. It's about using h(*) as the regularization, as I mentioned in the review. How gradients propagate to beta, through h(*) is not clear to me and this needs an analysis.
> >
> > (4) Discussing related references is appreciated.
> >
> > (5) Since there is an EMA step and the authors introduce new variables (distillation decoder), how to exactly implement the method is not not clear to me. My suggestion on providing 'a training loop with a formal description of EMA' is ignored.
> >
> > (6) Echoing the last bullet point, my comment 'no code is promised or released' is also ignored. This may be fine for technically clear manuscripts but since this one lacks in clarity, this becomes an issue.
> >
> > (7) BTW, what is 'Weakness-Inference'? Does it mean 'Weakness-Reference'?
> >
> > (8) I am still not convinced by this significance claim. Since I do not work on ImageNet pre-training myself, this is left for others to judge. BTW, to be honest, I am not convinced by the significance of SdAE too. What's worse, that paper's github repo is still empty, which clearly violates the policy of this year's ECCV. So using that paper as a supporting material is not convincing to me.
> >
> > (9) My comment on 'gradual reconstructed vicinity' is ignored, and I insist that this method does not contain technically sound 'gradual reconstructed vicinity'.
> >
> > (10) I am open-minded to discussion but not convinced by the 'local prior in late transformer layers'. I believe the papers in related works part 1 only empirically show the impact of locality in early layers but not in transformers. 'local prior is of vital importance for ViTs' is not clear to me. Meanwhile, the 'conjecture' is not convincing to me, there is no technically sound 'gradual reconstructed vicinity' and I don't understand what it exactly means.
> >
> > (11) Addressing the 'diverse' issue is appreciated. This enhances the rigor.
> >
> > (12) Clarifying about the calculation of distance map is appreciated, but this negligible claim is not supported with wall-clock time analysis. 'negligible' is a relative thing.
> >
> > (13) Repharasing about convergence is appreciated. This enhances the rigor. But it seems that 'As a tokenizer-free framework, our DDAE can not only achieve higher performance, but also accelerate the convergence speed of pretraining' is not revised.
> >
> > (14) My writing suggestion on 'The Experimental results' -> 'The experimental results' is ignored. I am not a native speaker and if the authors insist that this is the right grammar, please provide some references.
> >
> > I am open-minded to more rounds of discussion. But to me, the clarity, rigor and significance is still not satisfactory. So currently my recommendation remains as BR.

---

> > > ### Author Response · Authors · 2022-11-14
> > > **Response to Response**
> > >
> > > Dear Reviewer x4N6:
> > >
> > > Thanks for your response, here is our further reply.
> > >
> > > (3) We would like to clarify that **$h$ is not for regularization**, but a function for generating dynamic loss mask from distance map, as **defined in equation 2**. We use $L_2$ regularization here to avoid the extreme value of beta (in which way the model only focus on the hardest or simplest patches). **Beta is updated through normal backpropagation. Beta’s all relevant calculation formulas are clearly defined in the paper.** We noticed that your original comment was that “Since beta can be either negative or positive, regularizing its L1-norm is not valid. So I can see the authors use h for regularization, but the behavior is not readily clear to me.” We believe you have some misunderstandings here. As for the dynamic changing curve of beta, you can refer to Fig 5 and Fig 6.
> > >
> > > (5) In this paper the EMA step has nothing to do with the decoder, it only exists between the encoder and the momentum encoder. We have added the formal description of EMA in equation 4 as suggested, please see the revised paper.
> > >
> > > (6) If you have any additional clarifications, please do not hesitate to let us know. We will release the code in the future.
> > >
> > > (7) Modified.
> > >
> > > (8) Since you are not convinced of SdAE, we directly compare with MAE. CVPR 2022 paper MAE [1] outperforms ICLR 2022 paper BEiT [2] by +0.2% gains with 800epoch pretraining.  Our DDAE outperforms MAE with a significant margin in downstream finetuning. For classification on Imagenet, 100 epochs pretrained DDAE achieves 83.5% top-1 accuracy, surpassing 800 epochs pretrained MAE (83.4%); 800 epochs pretrained DDAE gets 84.3%, surpassing 1600 epochs pretrained MAE (83.6%) by a large margin (+0.7% gains). For semantic segmentation on ADE20K, DDAE achieves 49.3% mIoU for 800 epochs pretraining, surpassing MAE’s 48.1% mIoU with 1600 epoch pretraining largely (+1.2% gains). The detailed contrast is in Table 1 and 5.
> > >
> > > [1] Masked Autoencoders Are Scalable Vision Learners. CVPR 2022 oral.
> > > [2] BEiT: BERT Pre-Training of Image Transformers. ICLR 2022 oral.
> > >
> > > (9)(10) As for the locality for transformers, **all the papers in related works part 1 are based on vision transformers**, these works utilized different ways to inject local prior into transformer (distilled with CNN, using local attention mechanism or integrating convolution and transformer.) Our method empirically provides more local priors for the deep layers of ViTs as shown in Fig 4.
> > >
> > > We have revised the sentence”aided by the gradual reconstructed vicinity” to “aided by the simpler patches, which have been attached more importance in the earlier pre-training phase.” to avoid confusion.
> > >
> > > (12) The calculation of distance map only brings 1.6% extra wall-clock time cost compared with original data-processing. Furthermore, since the generation of distance map is in data-processing, this can be further accelerated simply with more workers used for parallel processing.
> > >
> > > (13) Modified to “As a tokenizer-free framework, our DDAE acheives consistent advantages both in a short schedule and a longer schedule.”
> > >
> > > (14) Modified.
> > >
> > > We have also revised the response to all reviewers above.

---

> > > > ### Comment · Reviewer_x4N6 · 2022-11-14
> > > > **Response**
> > > >
> > > > (3) Sorry I cannot understand this. According to Equation.6, the regularization is imposed on h(m). So I think the gradients are propagated through h(*)? And what I ask is exactly an analysis why doing this can 'avoid the extreme value of beta'.
> > > >
> > > > (5) I cannot understand this too. It seems, in the figure, that the EMA-ed encoder's output is used as the target for the distillation decoder. So, why it 'has nothing to do with the decoder'? The EMA step changes the regression target of the distillation decoder. Since there are multiple losses and an EMA step, the order between them should be noted in a complete training loop, for better reproducibility.
> > > >
> > > > (6) The code promise is appreciated. And I am looking forward to it, if accepted.
> > > >
> > > > (12) I think in MAE, the masked tokens are randomly selected in each iteration. How could we pre-process this since the distance transform should change from iteration to iteration.

---

> > > > > ### Author Response · Authors · 2022-11-15
> > > > > **Further Response**
> > > > >
> > > > > Dear Reviewer x4N6:
> > > > >
> > > > > Thanks for your response, here is our further response.
> > > > >
> > > > > (3) We got why you are confused. Yes, the gradients are propagated through $h$. Here $h$ is actually a softmax function of (the masked values of (the beta power of the distance map)). Let’s consider the output of softmax, **$L_2$ norm loss will force the output of softmax to be a uniform distribution.** For example, for a sequence $A$ of four elements [0.1, 0.2, 0.3, 0.4] (the sum is 1 since they are outputs of softmax), the $L_2$ norm loss will enforce $A$ to be [0.25,0.25,0.25,0.25], in this way its $L_2$ norm is minimal and $L_2$ norm loss is minimal. In this uniform distribution, the loss weight for all patches are the same. So this $L_2$ norm term is to ensure that the loss mask does not deviate too far from a uniform mask, this can effectively prevent beta from being too extreme as shown in our experiments.
> > > > >
> > > > > (5) We got why you are confused. The response "has nothing to do with the decoder" is not accurate and here is our further clarificaiton: The parameters of the momentum encoder are not updated by gradients (they do not have gradients, in pytorch is: require_grads=False), they are updated through equation 4. The outputs of the momentum encoder simply serve as targets, the same role as labels in classification task. **As for the order of updating, all losses are summed up to backward together, then the encoder and two decoders are updated through normal backpropogation. After this, the momentum encoder is updated through equation 4.** This is a common practice in self-supervised learning, and we recommend MoCo [1] and BYOL [2] for more description of the momentum encoder. Also, you can refer to the codebase of BootMAE [3] for detailed implementation, which is exactly where we build codebase from. Moreover, we have added some descriptions to make equation 4 clearer to those unfamiliar with this field.
> > > > >
> > > > > [1] Momentum Contrast for Unsupervised Visual Representation Learning. CVPR 2020.
> > > > > [2] Bootstrap Your Own Latent A New Approach to Self-Supervised Learning. NeurIPS2020.
> > > > > [3] Bootstrapped Masked Autoencoders for Vision BERT Pretraining. ECCV 2022.
> > > > >
> > > > >
> > > > > (12) Yes, the masked tokens are randomly selected in each iteration. But what we need is just a random mask right(14x14)? The generation of mask can be done in the data-loader, **we sincerely recommend the dataset.py** [1] of BootMAE. The calculation of distance map only depends on the mask, so we also define it in the function of data-loader. Since data-loader usually uses multiple workers to process data in parallel, the generation of masks and the calculation of distance maps can be greatly accelerated.
> > > > >
> > > > > [1]  BootMAE/datasets.py at main · LightDXY/BootMAE (github.com)
> > > > >
> > > > > Sincerely, Authors

---

> ### Author Response · Authors · 2022-11-23
> **About your last remaining suggestions**
>
> Dear Reviewer X4N6:
>
> **Clarity for Regularization**：We would like to clarify that we have added the explanation for why L2 norm is valid in the revised version, which is “Since L2 norm reaches minimum under uniform distributions, the L2 regularization loss here is to ensure that the loss mask does not deviate too far from a uniform mask, this can effectively prevent beta from being too extreme”.
>
> As for your request for gradient analysis,  beta’s all relevant calculation formulas are clearly defined in the paper and it is updated through normal backpropagation as we have highlighted. We can not understand why gradient analysis is recommended and what are we expected to analyze. The absolute value of gradient or the changing trend? We can not understand what’s the point **since we have already empirically shown that our regularization is effective, and explained why it is effective from a mathematical point of view**.
>
> **Training Loop**:  As we have stressed, the momentum encoder is a common practice in SSL, MoCo gives a detailed description of the algorithm because it is a pioneering work of using momentum encoder in SSL. For the current SSL works, **there is no training loop in recent works [1,2,3,4] that use the momentum encoders. Since we have given a necessary and concise description of the momentum encoder, and cited references that have detailed descriptions of momentum encoder for those unfamiliar with this field, we believe that this is clear enough both for those who are specialized in SSL and readers who are not familiar with this field.**
>
> We are grateful for your sincere suggestions. We have adopted many valuable suggestions from you, which helped us improve the quality and clarity.  For a few of your comments left, we gave detailed reasons why we didn't adopt them.  We have tried our best to answer every question you have raised. If there are omissions or you have more questions, please feel free to let us know.
>
> [1] SdAE: Self-distillated Masked Autoencoder. ECCV 2022.
> [2] Bootstrapped Masked Autoencoders for Vision BERT Pretraining. ECCV 2022.
> [3] MaskCLIP: Masked Self-Distillation Advances Contrastive Language-Image Pretraining[J]. arXiv preprint arXiv:2208.12262, 2022.
> [4] Exploring target representations for masked autoencoders[J]. arXiv preprint arXiv:2209.03917, 2022.

---

### Official Review · Reviewer_aYDP · 2022-11-03

**Confidence:** 3
**Correctness:** 2
**Technical Novelty And Significance:** 3
**Empirical Novelty And Significance:** 3
**Recommendation:** 5

**Clarity, Quality, Novelty And Reproducibility:**

As mentioned above in Cons, this paper is not well-written and hard to follow. The basic idea of providing more local priors to MIM is interesting, but the design of the method is not satisfactory.

**Strength And Weaknesses:**

Pros:

- Interesting exploration to improving MIM by providing more local priors.
- State-of-the-art performance among tokenizer-free methods.

Cons:

- The motivation of this paper is not convincing. The reconstruction difficulty of patches are not necessarily correlated with distance from visible patches. For example, it may depend on how much information is included in the patch. The background patches will still be easy to reconstruct following regular patterns even if it is far away from visible patches, but the foreground patches (e.g., nose of a dog) are hard to guess and require stronger semantic reasoning.

- The empirical improvement is too marginal. This is no clear advantage of the proposed method compared to SdAE(Chen et al., 2022c) according to the experiments. It has the same accuracy as SdAE for 100 epochs and only outperforms SdAE by 0.3% for 800 epochs.

- This paper is not well-written. The introduction is too redundant and messy. Figure 1 is hard to follow. The caption is not self-contained.

**Summary Of The Paper:**

This paper introduces a new MIM framework that dynamically focuses on patch reconstructions based on the degree of difficulty (i.e., the nearby visible patches) during pre-training. Besides, this paper proposes to self-distill intermediate features from the momentum encoder. Experiments show that it outperforms previous self-supervised methods on ImageNet for image classification and on ADE20K for semantic segmentation.

**Summary Of The Review:**

Overall, I think this paper is a borderline paper and slightly below the acceptance threshold. I would like the authors to provide more justifications to their motivation to further judge this paper.

---

> ### Author Response · Authors · 2022-11-08
> **Response to Reviewer aYDP**
>
> We sincerely thank the reviewer for the constructive comments. We have revised the paper accordingly (please see the revised version) and provided our detailed response below.
>
> **Concern-motivation：**
> 	Thank you for raising this question. Although not completely irrelevant, we agree that the reconstruction difficulty of background patches is less relevant with distance from visible patches. We have made justification to the expression of motivation in the revised paper and mentioned this problem in the conclusion and limitations part.
>
> In the case of random mask generation, the probability of foreground background distribution in visible and invisible tokens is uniform. The difference of distance we discussed does not conflict with the difference of background and foreground. At least deep dynamic supervision is harmless for background patches but beneficial for foreground patches.   **Harmless for background patches：** The mathematical expectation of the sum of loss weights on background patches will not change since mask is randomly generated, thus deep dynamic supervision will not make the model pay more attention to the background.   **Beneficial for foreground patches：** On the other hand，our designs do make sense to reconstruct foreground in a more reasonable way， which is supported by the experimental results and appreciated by reviewer pPUj (makes a lot sense), reviewer x4N6 (novel) and reviewer A9rs (gives a strong motivation).
>
>
> Existing MIM methods treat all patches equally regardless of **difficulty** (corresponding to the distance from visible patches in this work) and **importance** (the foreground and background difference), we focus on the first problem and propose a new framework supported by effective experiments. The latter problem also makes sense and the advantages brought by these two aspects shall be stackable, but how to distinguish foreground and background in an unsupervised way and design beneficial corresponding mechanism is beyond this work’s scope and we would like to explore it in the future work. Thanks again for pointing out this.
>
> **Concern-significance:**
> 	 Firstly, our core contribution is to point out an important problem ignored by the current MIM framework: reconstructing all patches equivalently is obviously suboptimal. We propose deep dynamic supervision to solve this problem and have achieved effective results. Our discussion and framework on this problem opened the door to explore a more reasonable and dynamic pretraining mechanism for MIM.
>
> Moreover, we would like to clarify that +0.3% top-1 accuracy gains over ECCV 2022 SOTA SdAE [1] on ImageNet without extra tokenizer is not marginal. For instance, SdAE surpasses CVPR 2022 paper MAE [2] with +0.4% gains with 800 epochs pretraining, and  MAE outperforms ICLR 2022 paper BEiT [3] by +0.2% gains with 800 epochs pretraining.
>
> When directly compared with MAE, our DDAE outperforms MAE with a significant margin in downstream finetuning. For classification on Imagenet, 100 epochs pretrained DDAE achieves 83.5% top-1 accuracy, surpassing 800 epochs pretrained MAE (83.4%); 800 epochs pretrained DDAE gets 84.3%, surpassing 1600 epochs pretrained MAE (83.6%) by a large margin (+0.7% gains). For semantic segmentation on ADE20K, DDAE achieves 49.3% mIoU for 800 epochs pretraining, surpassing MAE’s 48.1% mIoU with 1600 epoch pretraining significantly (+1.2% gains). The detailed contrast is in Table 1 and 5.
>
> [1] SdAE: Self-distillated Masked Autoencoder. ECCV 2022.
> [2] Masked Autoencoders Are Scalable Vision Learners. CVPR 2022 oral.
> [3] BEiT: BERT Pre-Training of Image Transformers. ICLR 2022 oral.
>
> **Concern-writing:**
> 	Thank you for your constructive suggestions, we have revised the introduction as suggested. We have rewritten the caption in Figure 1 to make it clear, please see the revised paper.
>
> Sincerely,
>
> Authors

---

> ### Author Response · Authors · 2022-11-16
> **Looking Forward to Your Reply**
>
> Dear Reviewer aYDP：
>
> Thank you again for the suggestions, which have greatly helped us improve the paper.
>
> We have revised the paper as suggested (please see the revised version) and responsed to your concerns and questions as the response below, but have not heard your post-rebuttal responses yet.
>
> Please do not hesitate to let us know if you have any additional clarifications or experiments that we can offer, as we hope to have an effective discussion with you to address your concerns and to convince you of the merits of our work.
>
> Sincerely, Authors

---

### Official Review · Reviewer_pPUj · 2022-11-03

**Confidence:** 4
**Correctness:** 3
**Technical Novelty And Significance:** 2
**Empirical Novelty And Significance:** 3
**Recommendation:** 5

**Clarity, Quality, Novelty And Reproducibility:**

This paper proposes a very simple modification to the MIM framework. The intuition behind the approach is clear, and the experimental results are adequate. This paper is well-written in general and it should be not difficult to reproduce the results.



**Strength And Weaknesses:**

Strength:
- The intuition that reconstructing patches has different levels of difficulty makes a lot of sense. The proposed method is also simple, and the required modifications on top of the MAE framework are quite minimal.
- The experimental results are solid.
- The analysis of locality(eg. Fig 4) is interesting and inspiring.

Weakness:
- The author claims that deep supervision introduces diversification and mentions the independence of beta_i. However, the Fig 5 shows that the beta_i are highly correlated and mostly the same. The difference in beta_i as shown in zoom-in figure in Fig 5 right plot seems too brittle to support the hypothesis that lower layer focus on more simple patches and deeper layers focus on more difficult patches. Also, why is the beta curve for block 12 missing from Fig 5?
- More discussion of how the initial beta is chosen is needed.  The paper initialize beta to -0.5 and the during the whole training beta varies from -0.6 to -0.425, very close to the initial value. What will happen if you initialize beta differently?


Other comments


**Summary Of The Paper:**

This paper introduce a simple mechanism to allow the training of MIM model tacking the reconstruction difficulty into account. It also introduce deep supervision that can help the diversify the feature representation across different layers. Good performance are reported to justfiy its methods.

**Summary Of The Review:**

The presented approach is simple yet effective as shown in their benchmark results. The analysis is in general good. Thus I am leaning to accept.

Post rebuttal:
I decided to down-rate the score after the reviewer discussion. Although the core idea of weighting patch reconstruction is inspiring, the improvement from dynamic loss is comparably marginal compared to the deep supervision, which makes the current paper writing less proper as the majority discussion is focused on dynamic loss. Given the current reviewers opinions, I recommend the authors to rephrase the paper writing or provide stronger results to validate dynamic loss and submit to another venue.

---

> ### Author Response · Authors · 2022-11-08
> **Response to Reviewer pPUj**
>
> We sincerely thank the reviewer for the constructive comments. We have revised the paper accordingly (please see the revised version) and provided our detailed response below.
>
> **Weakness 1：**
> 	Thank you for your valuable question. We agree that the claim for diversification is not strongly supported by the experiments. Since dropping this claim does not diminish our contributions for proposing dynamic reconstruction mechanism in MIM and migrating deep supervision in an effective way, we do not claim for diversification anymore in the revised version to make the paper more rigorous. As for block 12, the same trend is reflected as block 3,6,9(drops sharply at the beginning and rises slowly as the training progresses). Since our framework differs from others mainly in deep dynamic supervision for intermediate layers, we focus on the betas of block 3,6,9.
>
> **Weakness 2：**
> 	The final values of beta and the relative relationship among blocks are robust to initialization. We further depict the beta curve of different initialization in **Appendix B** in the revised paper, the similar trend is reflected. Interestingly, the final values of beta and the relative relationship among blocks tend to be stable in Fig 6. The final relative relationship is unchanged regardless of the **reverse initialization**. We initialize beta close to the final values for simplicity and better performance as shown in Table 7. Please see the revised paper.
>
> Sincerely,
>
> Authors

---

### Author Response · Authors · 2022-11-08
**Response to all reviewers**

We would like to thank the reviewers for their feedback and help in improving our work. We are happy that they found our works gives a strong motivation (A9rs), intuition makes a lot sense (pPUj) and interesting (aYDP), the experiments are solid (pPUj, A9rs) and detailed (VeHD), and appreciate the analysis (pPUj). Below we have responsed to the individual concerns and answered the questions of the reviewers. We have also revised the paper as suggested (the modifications are highlighted with blue color).

---

> ### Comment · Reviewer_x4N6 · 2022-11-14
> **Response**
>
> I think the method is novel in concept. But my original comment about the analysis is 'The attempt to reveal the underlying mechanism is appreciated (fig.3 and 4), although I am not fully convinced.' Meanwhile, since many of my suggestions are ignored, currently I don't buy this 'have addressed the individual concerns'.

---

> > ### Author Response · Authors · 2022-11-14
> > **Further Response**
> >
> > Thank you for your detailed reply. We have further replied to your remaining questions and we hope these further responses can eliminate your remaining concerns, please see the response to response. https://openreview.net/forum?id=k4p382L0bw&noteId=8wBGuS8YU1B
> >
> > #update
> > Thank you for the multiple rounds discussion with us，and thank you for raising the score！
> >
> > Sincerely, Authors

---

### Author Response · Authors · 2022-11-11
**Looking Forward to Your Feedback**

Dear Reviewers,

Thank you again for the suggestions, which have greatly helped us improve the paper.

We have revised the paper as suggested (please see the revised version) but have not heard any post-rebuttal responses yet.

Please do not hesitate to let us know if you have any additional clarifications or experiments that we can offer, as we would love to convince you of the merits of our work.

Sincerely, Authors

---

### Decision · Program_Chairs · 2023-01-20

**Decision:**

Reject

**Justification For Why Not Higher Score:**

Five out of six reviewers were negative about the paper, and the only positive reviewer did not find the rebuttal convincing.

**Justification For Why Not Lower Score:**

Lowest already

**Metareview: Summary, Strengths And Weaknesses:**

Five out of six reviewers were negative about the paper, and the only positive reviewer did not find the rebuttal convincing. Hence, the decision is **not** to recommend acceptance.